# The Effects of Different Reheating Methods on the Quality of Pre-Cooked Braised Chicken

**DOI:** 10.3390/foods14050868

**Published:** 2025-03-03

**Authors:** Sihao Liu, Yu Wang, Hewei Shi, Huijuan Zhao, Jiansheng Zhao, Shaohua Meng, Shifeng Shen, Junguang Li

**Affiliations:** 1Food Laboratory of Zhongyuan, Zhengzhou University of Light Industry, Luohe 462300, China; lsio642@163.com (S.L.); wy92@zzuli.edu.cn (Y.W.); h04116666@163.com (H.S.); zhaohuijuan@doyoo.cn (H.Z.); shenshifeng@zyfoodlab.com (S.S.); 2College of Food and Bioengineering, Zhengzhou University of Light Industry, Zhengzhou 450001, China; 3Henan Shuanghui Investment and Development Co., Ltd., Luohe 462000, China; zjs4567@163.com (J.Z.); mshua@163.com (S.M.)

**Keywords:** pre-cooked braised chicken, reheating methods, edible quality, pre-cooked dishes

## Abstract

This study investigated the effects of microwave (MW) reheating, water boiling (WB) reheating, and steaming (ST) reheating on the quality attributes (including reheating loss, moisture content, centrifugal loss, water distribution, color, texture, microstructure, flavor, and taste) of pre-cooked braised chicken (PBC), using a non-reheated group as a control (C). The results showed that the ST group demonstrated the lowest reheating loss, and that ST reheating had the least influence on textural characteristics among all the reheating methods. In addition, the results of the scanning electron microscope (SEM) showed that the integrity of the muscle fibers in the ST group was most comparable to the C group. Meanwhile, the aroma of the ST group was similar to that of the other reheating groups, and it exhibited a greater taste intensity. The ST treatment emerges as a viable reheating method for preserving the quality characteristics of PBC.

## 1. Introduction

Compared to home-cooked dishes, pre-cooked dishes have gained increasing popularity among consumers due to their greater convenience and shorter preparation time in the past decade. As the food industry continues to evolve, pre-cooked dishes are poised to dominate the future food retail market [1]. Braised chicken is a renowned traditional Chinese dish that primarily utilizes chicken drumsticks, complemented by mushrooms and a variety of spices to enhance freshness. The emergence of pre-cooked braised chicken (PBC) signifies the transformation of conventional dishes in response to contemporary industrialization.

Although PBC is classified as a cooked meat product, it typically requires reheating to a sensorily acceptable temperature before consumption as a pre-cooked dish. However, numerous studies primarily focus on the initial heat treatment of raw meat, regarding the selected thermal processing methods’ effect on the quality of meat. Wang et al. found that MW heating at 800 W for 100 s and 400 W for 200 s was more capable of rapidly forming a crust on the meat surface and reducing cooking losses compared to conventional heating [2]. Picouet et al. showed that MW heating at a power level of 798 W for 180 s results in accelerated heating and greater energy expenditure as the fat content in meat products rises [3]. Shi et al. found that ST heating was more effective in increasing the content of volatile flavor compounds under different heating methods [4]. The research conducted by Danowska et al. revealed that turkey meat exhibits better quality when subjected to ST heating at 220 °C with a saturation level of 90% [5]. Obviously, the final food quality, including physicochemical and sensory properties, varies according to different heating treatments and their different principles of heat transfer [6]. Nevertheless, it remains to be determined whether the same conclusions can be applied to cooked meat products. Research has mainly neglected the impact of reheating treatments on pre-cooked dishes.

Therefore, the purpose of this study is to explore the effects of different reheating methods on the quality characteristics of PBC, thus providing useful information in the selection of appropriate reheating methods for the consumption of PBC. The findings may also be applicable to the reheating of other pre-cooked meat products. Microwave (MW) reheating, water boiling (WB) reheating, and steaming (ST) reheating were selected for this study due to their universality in both commercial and domestic environments [7].

## 2. Materials and Methods

### 2.1. Materials and Reagents

PBCs (with a 2-month shelf life) were provided by Henan Shuanghui Investment and Development Co., Ltd. (Luohe, China). Glutaraldehyde reagent was purchased Beijing Solarbio Science & Technology Co., Ltd. (Beijing, China).

### 2.2. Reheating Process

The samples were divided into four groups for different reheating treatments, in which group C was the control group without a reheating treatment; the MW group included sampled that were reheated with a microwave oven with a power of 800 W for 2 min (PC20M5W, Midea, Foshan, China); the WB group included samples that were reheated by directly placing the original vacuum packaging bag of PBC into boiling water at standard atmospheric pressure for 8 min; and the ST group included samples that were reheated using an electric steamer at 100 °C for 14 min (ZGC322301, Midea, China). All reheating group samples were measured using a multiplexed temperature checker (AT4508, Anbai, Changzhou, China). When the temperature in center of the meat reached 72.5 °C [8], the reheating treatment was ended immediately.

### 2.3. Reheating Loss

The mass of each sample was accurately measured using an electronic analytical balance (FA124X, D&T, Tianjin, China) before reheating, denoted as m_i_. After the reheating process was completed, the samples were allowed to cool at room temperature for 30 min. Any excess moisture on the surface was removed using a paper towel, and the mass at this point was measured and recorded as m_f_. The formula for calculating the reheating loss was as follows:(1)$$Reheating\;Loss\left(\%\right)=\frac{m_{i}-m_{f}}{m_{i}}\times 100$$

### 2.4. Moisture Content

Upon completion of the reheating process, the samples from each group were permitted to equilibrate at room temperature for 30 min. Any excess moisture on the surface was eliminated using absorbent tissue, and a sample weighing 5 ± 0.5 g was measured (with the mass recorded at this stage as m_r_). The sample was subsequently subjected to drying at a temperature of 105 ± 2 °C until a stable weight was attained. Following the drying phase, the mass was documented as m_d_. The equation for determining the moisture content was as follows:(2)$$Moisture\;Content(\%)=\frac{m_{r}-m_{d}}{m_{r}}\times 100$$

### 2.5. Centrifugal Loss

After the completion of reheating, the samples from each group were allowed to cool at room temperature for 30 min. Excess moisture on the surface was removed using tissue paper, and a sample of 10 ± 0.5 g was weighed (recording the weight at this time as m_1_) and placed in a 50 mL centrifuge tube lined with filter paper. The tube was then centrifuged at 10,000 rpm for 10 min. After centrifugation, the filter paper was opened, and the remaining mass of the sample was weighed and recorded as m_2_. The formula for calculating the water-holding capacity was as follows:(3)$$Centrifugal\;loss\left(\%\right)=(1-\frac{m_{2}}{m_{1}})\times 100$$

### 2.6. Low Field Nuclear Magnetic Resonance (LF-NMR)

Each sample was adjusted to 10 mm × 10 mm × 10 mm in size and placed in a 15 mm diameter cylindrical glass tube supplied with the instrument for testing. LF-NMR relaxation measurements were performed using the previously reported method [9] with an 18 MHz LF-NMR analyzer (NM120, NIUMAG, Suzhou, China). The measurement was conducted at a temperature of 32 ± 0.01 °C, with a proton resonance frequency of 18 MHz. The transverse relaxation time T_2_ was assessed using the Carr–Purcell–Meiboom–Gill sequence, where the half-echo time value τ (the interval between the 90° and 180° pulses) was set at 250 μs with 32 scans and 12,000 echoes.

### 2.7. Color Determination

The color composition of each sample (*L**, *a**, *b**) was measured using a colorimeter (RM200QC, Xrite, Grand Rapids, MI, USA), which must be calibrated prior to measurement. Samples were sectioned into cubes measuring 10 mm × 10 mm × 10 mm. *L** indicates brightness (with 100 representing pure white and 0 representing pure black), *a** denotes the red–green axis (where positive values indicate red and negative values indicate green), and *b** represents the yellow–blue axis (with positive values indicating yellow and negative values indicating blue). The formula for the total color difference ΔE was calculated as follows:(4)$$\Delta E=\sqrt{{\left(\Delta L*\right)}^{2}+{\left(\Delta a*\right)}^{2}+{\left(\Delta b*\right)}^{2}}$$

### 2.8. Texture Profile Analysis

Each sample was cooled at room temperature for 3 h and trimmed into 8 mm × 8 mm × 8 mm tissue blocks. The texture profile analysis was appropriately adjusted and conducted according to the method of Ruiz with a texture analyzer (CTX, Brookfield, MA, USA) [9,10]. All tests were performed in TPA mode with a TA-AACC 36 probe. The test velocity was 1 mm/s, and two consecutive measurements were taken with a profile variable of 50%.

### 2.9. Scanning Electron Microscope

In accordance with the methodology established by Zhou [11], the samples from each group were sectioned into tissue blocks measuring 8 mm × 8 mm × 2 mm, which were subsequently fixed in pre-cooled 2.5% glutaraldehyde at 4 °C for a duration of 24 h. Following fixation, the samples underwent a series of elutions with 10%, 30%, 50%, 70%, 80%, 90%, and 100% ethanol for 10 min each. After undergoing freeze-drying (12N/A, SCIENTZ, Ningbo, China), the samples were treated with gold spraying. The surface morphology of the transversal surface from the muscle fibers was examined using a scanning electron microscope (Nova Nano SEM 450, FEI, Hillsboro, OR, USA) at a magnification of 350×.

### 2.10. Electronic Nose Analysis (E-Nose)

Utilizing the detection method by Bai [12], 5 g of the samples from each group was placed into a 20 mL sealed vial and incubated in a water bath at 25 °C for 25 min. Prior to analysis with the electronic nose (PEN3, AIRSENSE, Schwerin, Germany), the chamber was purged with clean air until baseline stability was achieved, followed by sample measurement. The measurement duration was set to 180 s, with data collected during the 120 s equilibrium phase for analysis.

### 2.11. Electronic Tongue Analysis (E-Tongue)

The detection method proposed by Yuan [13] was adopted and modified. A total of 50 g of the samples from each group was chopped and mixed with 150 mL of deionized water. This mixture was homogenized at 10,000 rpm for 10 s using a homogenizer (A25, Oulior, Shanghai, China), followed by filtration through four layers of gauze. The homogenization and filtration steps were repeated twice. The resulting filtrate was equilibrated overnight at 4 °C in preparation for subsequent analysis. A taste analysis was conducted using an electronic tongue (cTongue, BosinTech, Shanghai, China), with a data collection duration of 120 s and a sampling interval of 1 s, without any collection delay. Each sample was analyzed 5 times, with the stable values from the middle 3 measurements being recorded as the test results.

### 2.12. Data Analysis

All experiments were repeated at least in triplicate. Experimental data were expressed as mean ± standard deviation. A one-way analysis of variance (ANOVA) was performed using SPSS 27.0 (IBM, Armonk, NY, USA); when the *p* value < 0.05, it was considered significant. GraphPad Prism 10.0.1 (GraphPad, La Jolla, CA, USA) was used for the graphical analysis. Origin Pro 2021 (OriginLab, Northampton, MA, USA) was selected for the PCA analysis.

## 3. Results and Discussion

### 3.1. Reheating Loss, Moisture Content, and Centrifugal Loss Analysis

As illustrated in Figure 1, all reheating treatment groups exhibited visible reheating loss. The reheating loss in the MW group was significantly greater than that in the WB and ST groups (*p* < 0.05). This was attributable to the thermal treatment causing the muscle fibers of the meat products to contract during the reheating process, which altered the intracellular structure and led to varying degrees of loss of moisture, soluble proteins, lipids, vitamins, and trace elements from the muscle tissue into the extracellular space [14]. Furthermore, a positive correlation existed between reheating loss and reheating intensity [15]. After reheating, the moisture content of samples across all reheating groups generally decreased, with the MW group exhibiting the highest moisture loss (*p* < 0.05), while the difference between the WB and ST groups was minimal. This trend was also observed in the study by Wang et al. [8]. Concurrently, post reheating, the centrifugal loss of the samples from the MW and WB groups was reduced. This phenomenon may be attributed to the effect of protein denaturation. Research by Franz et al. indicated that the gelation of myosin, sarcoplasmic proteins, or collagen can induce the formation of bound water, thereby enhancing the water-holding capacity after the reheating treatment [16,17]. Meanwhile, the MW group showed the most significant decline (*p* < 0.05). This is understandable, because in the reheating treatment, the predominant loss was attributed to moisture, which could be certified in terms of moisture content. Consequently, the amount of water that might be lost through centrifugation in the MW group was also reduced compared to the other groups. From the perspective of these three indicators, the quality of the ST group was relatively similar to that of the C group.

### 3.2. Evolution of Water Distribution

LF-NMR is a highly sensitive technique for detecting the overall moisture distribution in food products. Moisture in meat products can be categorized into three types based on how tightly the water is bound to the muscle components: bound water (T_2b_, 1–20 ms), immobilized water (T_21_, 10–100 ms), and free water (T_22_, 100–1000 ms) [18]. Figure 2 shows the evolution of the moisture distribution in PBC under various reheating conditions. Following the reheating process, the free water content in all reheated samples significantly decreased compared to the C group (*p* < 0.05), while the content of immobilized water markedly increased. The bound water content showed no significant change, indicating that during reheating, free water underwent bidirectional movement, which was caused by the thermal denaturation of proteins induced by the heat treatment. A portion of the free water is lost due to the transverse and longitudinal contraction of muscle fibers [19], while another portion transitions into immobilized water [20]. Additionally, it is worth noting that the T_22_ peak in the reheating treatment groups exhibited considerable shrinkage and a leftward shift. This suggested a reduction in the fluidity of free water. The primary factor contributing to this outcome was the oxidative denaturation of proteins within the samples, which led to alterations in the structure of muscle fibers. The ST group exhibited the highest content of immobile water compared to all other group, and the same finding was also found in the study of Lian et al. [21]. The free water content of the MW group was the highest among the reheating groups, suggesting that the migration of free water in the MW group was most restricted, which also indicated a greater extent of cross-linking within its internal microstructure.

### 3.3. Changes in Color

The color changes of reheated PBC are shown in Figure 3. The *L** values for the MW and WB groups significantly decreased (*p* < 0.05), with the MW group exhibiting the most pronounced reduction. There was no significant difference in *L** values between the ST group samples and the control group. Similar findings were reported by Dai et al. [22]. The heating medium might be the main influencing factor. In the ST group, efficient contact of the sample with the flowing steam induced more loss of brown pigments. In the MW and WB groups, the limitation of flow characteristics and duration of the heat transfer medium contributed to the accumulation of brown substances within the muscle fibers. The *a** values in the reheating groups showed a general decline, with no significant difference between the MW and control groups. Furthermore, the *b** values of the samples decreased overall after reheating, with the MW group showing the most significant reduction (*p* < 0.05). There are studies indicating that this was the result of the oxidative denaturation of globin during the reheating stage [23,24]. The changes in ΔE for the reheating groups were ranked as MW > WB > ST (*p* < 0.05). Overall, the ST treatment better maintained the original color of the PBC.

### 3.4. Texture Profile Analysis

The textural characteristics of meat products serve as critical indicators in sensory evaluation, with the quality of texture directly influencing consumer consumption experiences. As indicated in Table 1, the reheating process significantly alters the textural properties of PBC, with hardness, springiness, and chewiness showing notable changes. The indices showed significant enhancement across all reheat-treated groups (*p* < 0.05). This improvement was attributed to the denaturation of myosin following reheating, which resulted in the contraction of myofibers and a reduction in myogenic fiber spacing [25]. Furthermore, the textural characteristics of the reheated groups were observed to increase in the following order: MW > WB > ST. This pattern aligned with the centrifugal loss rate trends documented in prior research. Additionally, it was established that the textural attributes of meat products were significantly influenced by their chemical composition, rather than being solely dependent on the alterations in textural properties due to changes in muscle structure [26]. The reheating treatment did not significantly affect the cohesiveness of PBC, suggesting that different reheating methods do not have a substantial impact on this indicator. Research indicates that the cohesiveness of meat is primarily related to its lipid content [27]. Consequently, based on the texture profile analysis, the ST group was able to better maintain the original texture of PBC.

### 3.5. Scanning Electron Microscope

Figure 4 illustrates the structural characteristics of PBC subjected to various reheating methods under a magnification of 350×. The results indicate a significant disparity in the effects of different reheating treatments on the myofibrillar structure of the samples. The muscle microstructure of both the C and ST groups consisted of tightly arranged muscle fibers with small gaps, but with distinct boundaries between individual muscle fibers [28]. In the WB and MW groups, the microscopic structure of muscle exhibited significant disorganization at the surface of the myofibers, with a lack of clear stratification and indistinct boundaries between individual myofibers. Meanwhile, the severity classification in both groups was ranked as MW > WB. This was attributed to the high-intensity reheating treatment, which induced the denaturation of myofibrillar proteins and dissolution of connective tissue, resulting in a more compact structure [29]. The microstructure revealed that the MW, WB, and ST reheating treatments affected the PBC to different degrees, aligning with the observed trends in the texture profile analysis concerning muscle, as well as the moisture content and migration of free water discussed earlier across each group.

### 3.6. Electronic Nose Analysis

The E-nose is widely used in the food industry for volatile flavor analysis due to its objective, non-destructive characteristics, and due to the fact that it is shielded from the influence of subjective factors in sensory evaluation [30]. This device is equipped with ten sensors, making it suitable for the analysis of a wide range of volatile flavor substances. Specifically, the W1C sensor is sensitive to aromatic compounds; the W5S sensor detects nitrogen oxides; the W3C and W6S sensors are responsive to aromatic compounds and ammonia; the W5C sensor is attuned to short-chain alkanes and aromatic components; the W1S sensor is sensitive to methyl compounds; the W1W sensor detects sulfides; the W2S sensor is responsive to alcohols, aldehydes, and ketones; the W2W sensor is sensitive to organic sulfides; and the W3S sensor is attuned to long-chain alkanes [31]. Furthermore, the e-nose can perform principal component analysis (PCA), transforming numerous variables into characteristic variables. The resulting data can be projected in 2D or 3D coordinate systems, facilitating the visualization of high-dimensional data [32].

Figure 5 presents the principal component analysis of volatile flavor compounds in the PBC under various reheating conditions, as analyzed by an electronic nose, along with the distribution of responses from each sensor. In Figure 5a, the sum of total proportions from principal component PC1 and secondary component PC2 was 95.8%, indicating a high reliability of the sample information in representing the actual conditions. The data among samples within the same group exhibited a high degree of clustering, confirming the reproducibility and stability of the samples. The C group was dispersed and did not overlap with the reheated groups, highlighting a significant difference in the composition of volatile flavor compounds between the reheated groups and the un-reheated group. Notably, the ST group showed some overlap with the WB and MW groups, while there was no overlap between the WB and MW groups, suggesting that the MW and WB reheating methods exert different effects on the composition of volatile flavor compounds. Concerning the impact of volatile flavor compounds, the effect of ST reheating lay between the other two reheating treatments, with its volatile flavor compound composition being similar to that of the other two reheating treatments. The trends in compositional changes are shown in Figure 5b. Following the MW treatment, the release of all types of volatile flavor compounds was enhanced; the WB and ST treatments increased the release of aromatic compounds, ammonia, and long-chain alkanes, while the proportion of other compound types diminished. This could be attributed to the low boiling point of aromatic compounds, which readily evaporated upon heating. The increased release of ammonia and long-chain alkanes indicated that protein and fat oxidation had occurred as a result of the reheated treatment.

### 3.7. Electronic Tongue Analysis

The e-tongue can convert trace flavor chemical signals into electrical signals, providing a comprehensive representation of the flavor distribution in food products. As illustrated in Figure 6, the sum of total proportions from the first principal component (PC1), second principal component (PC2), and third principal component (PC3) was 98.9%, effectively characterizing the taste profile of PBC. Following the reheating treatment, all reheated groups exhibited a significantly higher flavor intensity compared to the control group, a finding corroborated by Zhao et al. [33] in their study of reheated roast chicken. Furthermore, there was no overlap among the reheated groups, indicating that varying degrees of reheating significantly influenced the permeation of salt and the release of nucleotides and amino acids. Similar results were reported by Li et al. [34], who found that different cooking methods applied to Tibetan pigs resulted in distinct flavor characteristics. Additionally, the proximity of the MW and C groups may be attributed to the shorter reheating duration associated with the MW treatment [35]. In contrast, the ST group was positioned furthest from the C group. The reason for this result is likely due to the flowing steam, which, owing to the strong mobility and osmotic pressure, facilitated the permeation of salt and the release of flavor compounds at high temperatures, resulting in the most significant changes in flavor composition and intensity for the ST group.

## 4. Conclusions

In this study, the three different reheating methods had a significant impact on the quality characteristics of PBC. The data indicated that the ST treatment resulted in the lowest reheating losses, minimal color differences, and least texture alteration compared to other reheating methods. The microstructure observations further revealed that the ST treatment caused minimal damage to the muscle fibers of PBC. Additionally, the ST group exhibited similar flavor profiles to the other reheating groups but demonstrated a higher taste intensity. These findings suggest that ST reheating results in better eating quality and flavor characteristics in PBC. This study provides valuable insights for consumers in selecting appropriate reheating methods for PBC, with potential applications for other pre-cooked meat products as well.

## Figures and Tables

**Figure 1 foods-14-00868-f001:**
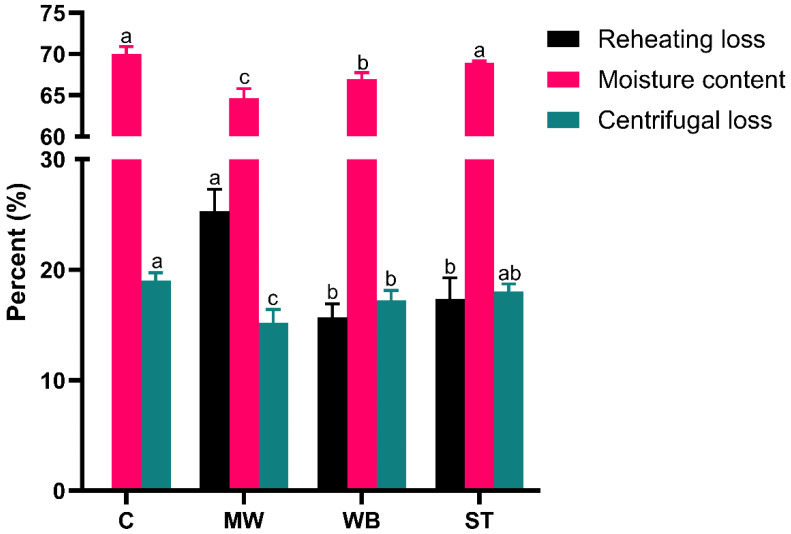
Effects of different reheating treatments on reheating loss, moisture content, and centrifugal loss of PBC. **Note:** C: the un-reheated samples, MW: the samples reheated using microwave, WB: the samples reheated using boiling water, ST: the samples reheated using steam. There are significant differences in different letters, *p* < 0.05.

**Figure 2 foods-14-00868-f002:**
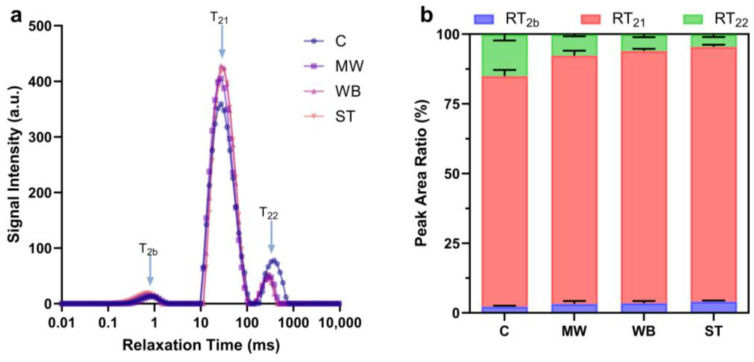
Evolution of water distribution of PBC under different reheating conditions. (**a**) The distribution curves of transverse relaxation time, (**b**) peak area ratio of T_2b_, T_21_, and T_22_. **Note:** C: the un-reheated samples, MW: the samples reheated using microwave, WB: the samples reheated using boiling water, ST: the samples reheated by steam.

**Figure 3 foods-14-00868-f003:**
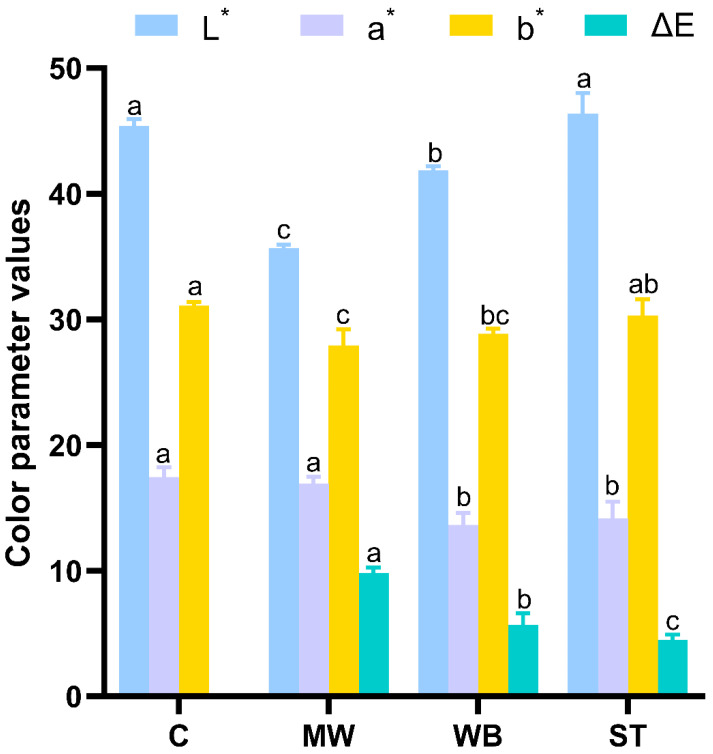
Effects of different reheating treatments on color of PBC. **Note:** C: the un-reheated samples, MW: the samples reheated using microwave, WB: the samples reheated using boiling water, ST: the samples reheated using steam. There are significant differences in different letters, *p* < 0.05.

**Figure 4 foods-14-00868-f004:**
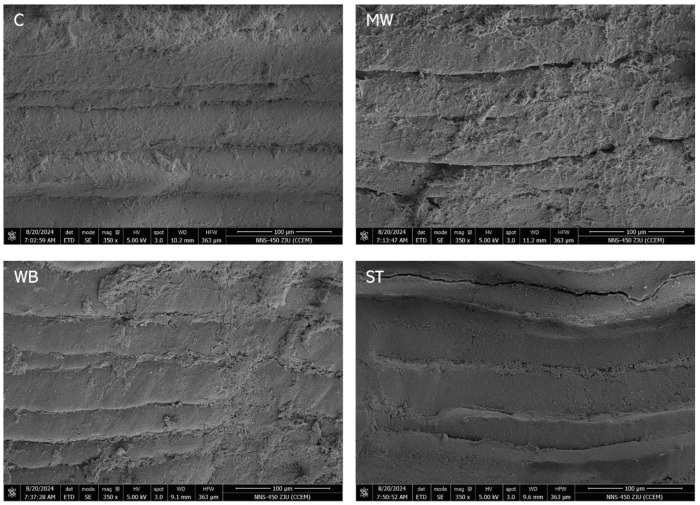
Scanning electron microscopy of PBC under a magnification of 350× after different reheating treatments. **Note:** C: the un-reheated samples, MW: the samples reheated using microwave, WB: the samples reheated using boiling water, ST: the samples reheated using steam.

**Figure 5 foods-14-00868-f005:**
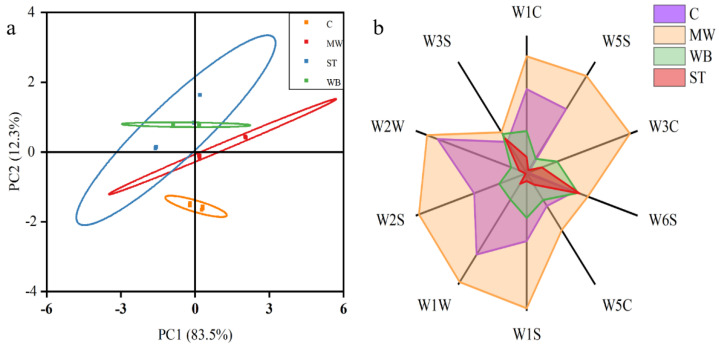
Effects of different reheating methods on volatile aroma compounds of PBC. (**a**) Principal component analysis, (**b**) relative response of each sensor at 120 s. **Note:** C: the un-reheated samples, MW: the samples reheated using microwave, WB: the samples reheated using boiling water, ST: the samples reheated using steam.

**Figure 6 foods-14-00868-f006:**
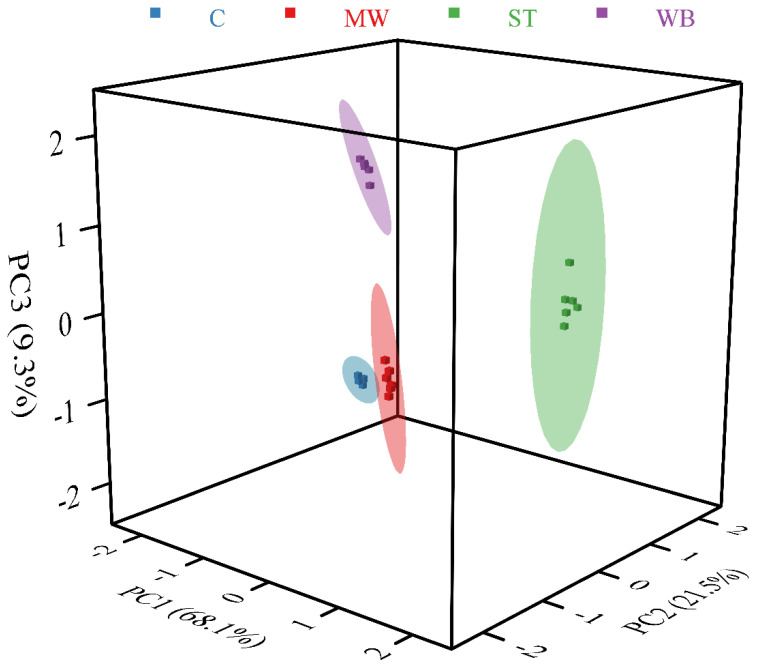
Effects of different reheating methods on the taste substances of PBC. **Note:** C: the un-reheated samples, MW: the samples reheated using microwave, WB: the samples reheated using boiling water, ST: the samples reheated using steam.

**Table 1 foods-14-00868-t001:** Effect of different reheating treatments on texture profile analysis of PBC.

Reheating Methods	Hardness (g)	Springiness (mm)	Chewiness (g · mm)	Cohesiveness
C	478.33 ± 65.58 ^c^	2.38 ± 0.05 ^c^	544.33 ± 47.5 ^d^	0.48 ± 0.06
MW	1096 ± 58.97 ^a^	2.96 ± 0.1 ^a^	1758.33 ± 53.14 ^a^	0.54 ± 0.02
WB	957.67 ± 81.95 ^a^	2.72 ± 0.08 ^b^	1469 ± 52.31 ^b^	0.57 ± 0.03
ST	714.67 ± 61.98 ^b^	2.58 ± 0.08 ^bc^	955.67 ± 80.31 ^c^	0.52 ± 0.04

**Note:** C: the un-reheated samples, MW: the samples reheated using microwave, WB: the samples reheated using boiling water, ST: the samples reheated using steam. Means in the same row with different letters differ significantly (*p* < 0.05).

## Data Availability

The original contributions presented in the study are included in the article, further inquiries can be directed to the corresponding author.

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
