# Peer review of "The Effects of Different Reheating Methods on the Quality of Pre-Cooked Braised Chicken"

_foods, 2025, doi:10.3390/foods14050868_

Round 1

Reviewer 1 Report

Comments and Suggestions for Authors

The paper "The effects of different reheating methods on the quality of pre-cooked braised chicken" presents an evaluation of cooking methods for a typical Chinese dish. The article is well designed and the results are relevant to food production. Some corrections need to be made. These corrections are indicated in the attached file.

Author Response

For research article

Response to Reviewer 1 Comments

1. Summary

Thank you very much for taking the time to review this manuscript. Please find the detailed responses below and the corresponding revision in the re-submitted files.

 2. Point-by-point response to Comments and Suggestions for Authors

Comments 1: The paper "The effects of different reheating methods on the quality of pre-cooked braised chicken" presents an evaluation of cooking methods for a typical Chinese dish. The article is well designed and the results are relevant to food production. Some corrections need to be made. These corrections are indicated in the attached file.

Response 1: Thank you for pointing this out. We agree with this comment. Therefore, we have made improvements to the comments provided, and the updated location is as follows:

  1. Regarding the opinions in Abstract, this section has been re-edited and updated in lines 12-21 of Re-manuscript-1.
  2. Regarding the opinions in Introduction, this section has been re-edited and updated in lines 34, 35, 37, 39 and 50-56 of Re-manuscript-1.
  3. Regarding the opinions in 2 Reheating process, this section has been re-edited and updated in lines 67 and 68 of Re-manuscript-1.
  4. Regarding the opinions in 5 Centrifugal loss, this section has been re-edited and updated in lines 88-91 of Re-manuscript-1.
  5. Regarding the opinions in 1 part from Results and discussion, this section has been re-edited and updated in lines 152-172 of Re-manuscript-1.

Reviewer 2 Report

Comments and Suggestions for Authors

1. The aim is precise and concerned with showing differences in the quality of pre-cooked braised chicken (PBC) using reheating methods like microwave treatment (MW), water boiling treatment (WB), and steaming treatment (ST).

2. Applied methods are correct and sufficiently described. Arouse questions why the variety of applied methods was not enriched by surveys concerning the most prevalent population methods of reheating pre-cooked braised chicken.

It needs to explain why your control sample is without reheating treatment. Does it mean that pre-cooked braised chicken (PBC) is edible, and does the panel compare the sensory properties of all cold group samples after different reheating times? There is a lack of panel analysis.

It is unclear if the control sample is comparable to other samples after reheating treatment.

In the Materials and Methods paragraph, you describe experiments only related to reheating samples. What about the control group?

3. The Results and Discussion paragraph presents the results obtained and discusses them with the results of other authors.

In Fig. 1, axis Y is or is not correctly presented; on axis X, we have in % moisture, reheating loss, and centrifugal loss values.

The same concerns Figure 2. Color parameters are along axes X and Y; we have values (X) of these parameters. Letters are better than stars for presenting significant differences.

4. Conclusions

What is the practical value of the experiment? What is the most preferable method for pre-cooked braised chicken reheating? Why is comparing different methods of product reheating essential? What problem is solved? Moreover, is it recommended for consumers from a nutritional perspective or to change their food preparation habits?

Therefore, the last statement (Nevertheless, the specific mechanisms by which various reheating methods influence the alterations in the physicochemical characteristics of pre-cooked meat dishes remain unclear, which necessitates further exploration) without reasonable reasons seems little practical.

Life preferences, lifestyle, nutritional value, and these elements are not included in this work to justify the purpose of the research conducted. What does this new work bring? Any innovative aspects.

5. The literature is thoroughly prepared and properly presented.

I appreciate the well-prepared manuscript, but it needs to underline the importance of the experiment and its practical meaning for consumers and scientists.

General small mistakes

Line 18, What does expression mean:  the highest fidelity

Line 35: slight mistake: . However, Numerous studies…

Line 88: some missing phrase?

Line 162: ST groups were reduced?

Author Response

For research article

Response to Reviewer 2 Comments

1. Summary

Thank you very much for taking the time to review this manuscript. Please find the detailed responses below and the corresponding revision in the re-submitted files.

 2. Point-by-point response to Comments and Suggestions for Authors

Comments 1: The aim is precise and concerned with showing differences in the quality of pre-cooked braised chicken (PBC) using reheating methods like microwave treatment (MW), water boiling treatment (WB), and steaming treatment (ST).

Response 1: Agree.

Comments 2: Applied methods are correct and sufficiently described. Arouse questions why the variety of applied methods was not enriched by surveys concerning the most prevalent population methods of reheating pre-cooked braised chicken.It needs to explain why your control sample is without reheating treatment. Does it mean that pre-cooked braised chicken (PBC) is edible, and does the panel compare the sensory properties of all cold group samples after different reheating times? There is a lack of panel analysis.It is unclear if the control sample is comparable to other samples after reheating treatment.In the Materials and Methods paragraph, you describe experiments only related to reheating samples. What about the control group?

Response 2: Thank you for pointing this out. Next, I will address this point in detail, step by step. 1st, concerning the limitations of reheating methods, our selection was based on user surveys, which led us to exclude less effective methods such as baking reheating and pan-frying reheating. As a result, these methods were not reflected in the research. 2nd, the control group did not undergo reheating treatment, as our aim was to investigate the specific effects of reheating on the quality characteristics of pre-cooked meat products (PBC). The changes in quality indicators before and after reheating are clearly demonstrated. Additionally, PBC is a cooked meat product that can be consumed directly without reheating, under cold conditions. However, to align with people's sensory preferences and eating habits, reheating is typically required. Regarding the sensory evaluation, we considered that PBC, as a prepared dish, might influence people's subjective assessments. Therefore, we opted to use instruments such as an electronic nose and electronic tongue to objectively analyze its sensory parameters. 3rd, the comparability between unheated and heated groups has been extensively studied in current research on raw meat. Therefore, we aimed to investigate whether similar conclusions could be applied to pre-made cooked meat products. 4th, the experimental method of the control group is consistent with the operation of the reheating group, this section has been re-edited and updated in lines 96, 105, 113, 119, 128 and 136 of Re-manuscript-1.

Comments 3: The Results and Discussion paragraph presents the results obtained and discusses them with the results of other authors. In Fig. 1, axis Y is or is not correctly presented; on axis X, we have in % moisture, reheating loss, and centrifugal loss values. The same concerns Figure 2. Color parameters are along axes X and Y; we have values (X) of these parameters. Letters are better than stars for presenting significant differences.

Response 3: Agree. We have updated Fig. 1 and Fig. 2 and used letters to distinguish significance. This section has been re-edited and updated in lines 174 and 223 of Re-manuscript-1.

Comments 4: Conclusions. What is the practical value of the experiment? What is the most preferable method for pre-cooked braised chicken reheating? Why is comparing different methods of product reheating essential? What problem is solved? Moreover, is it recommended for consumers from a nutritional perspective or to change their food preparation habits? Therefore, the last statement (Nevertheless, the specific mechanisms by which various reheating methods influence the alterations in the physicochemical characteristics of pre-cooked meat dishes remain unclear, which necessitates further exploration) without reasonable reasons seems little practical. Life preferences, lifestyle, nutritional value, and these elements are not included in this work to justify the purpose of the research conducted. What does this new work bring? Any innovative aspects.

Response 4: Thank you for pointing this out. Next, I will explain this point step by step. 1st, the practical value of the experiment is providing valuable insights for consumers in selecting appropriate reheating methods for PBC, with potential applications for other pre-cooked meat products as well. The most preferable method for PBC reheating is steaming treatment (ST). Numerous studies primarily focus on the initial heat treatment of raw meat regarding the quality of meat effected by the selected thermal processing methods. Nevertheless, it remains to be determined whether the same conclusions can be applied to cooked meat products. Mainly research neglects the impact of reheating treatment on pre-cooked dishes now. Moreover, this study recommended consumers to change their food preparation habits. 2nd, regarding the opinions in 4 Conclusion, this section has been re-edited and updated in lines 337-346 of Re-manuscript-1. 3rd, this experiment primarily serves as a supplement to address gaps in current related research.

Comments 5: The literature is thoroughly prepared and properly presented. I appreciate the well-prepared manuscript, but it needs to underline the importance of the experiment and its practical meaning for consumers and scientists. General small mistakes Line 18, What does expression mean: the highest fidelity. Line 35: slight mistake:  However, Numerous studies… Line 88: some missing phrase? Line 162: ST groups were reduced?

Response 5: Thank you for pointing this out. We agree with this comment. Therefore, we have made improvements to the comments provided, and the updated location is as follows:

  1. Regarding the opinions in Line 18, this section has been re-edited and updated in line 18 of Re-manuscript-1.
  2. Regarding the opinions in Line 35, this section has been re-edited and updated in line 35 of Re-manuscript-1.
  3. Regarding the opinions in Line 88, this section has been re-edited and updated in lines 88-91 of Re-manuscript-1.
  4. Regarding the opinions in Line 162, this section has been re-edited and updated in line 163 of Re-manuscript-1.

Round 2

Reviewer 2 Report

Comments and Suggestions for Authors

I agree or partly agree (ad.2) with the explanations given by the authors. The most important are listed undertaken by them:

  1. The control group did not undergo reheating treatment, as we aimed to investigate the effects of reheating on the quality characteristics of pre-cooked meat products (PBC).
  2. Regarding the sensory evaluation, we considered that PBC, as a prepared dish, might influence people's subjective assessments. Therefore, we used instruments such as an electronic nose and tongue to analyze its sensory parameters objectively. Comment: you can use GPT Chat to assess sensory evaluation or an electronic nose and tongue, but it is worth asking the panel for an assessment of curiosity. It means additional costs and time, but results could be better discussed. It is a suggestion for the future scientific work of the authors.
  3. The practical value of the experiment is that it provides valuable insights for consumers in selecting appropriate reheating methods for PBC, with potential applications for other pre-cooked meat products as well. The most preferable method for PBC reheating is steaming treatment (ST). Comment: the accents are correct for the importance of the experiment from a practical point of view.
  4. In response 4, I am not sure what the author means in a sentence: Mainly, research neglects the impact of reheating treatment on pre-cooked dishes now.

The last sentence in the conclusion paragraph is of great importance (This study provides valuable insights for consumers in selecting appropriate reheating methods for PBC, with potential applications for other pre-cooked meat products.

Author Response

For research article

Response to Reviewer 2 Comments

1. Summary

Thank you very much for taking the time to review this manuscript. Please find the detailed responses below and the corresponding revision in the re-submitted files.

 2. Point-by-point response to Comments and Suggestions for Authors
Comments

1: The control group did not undergo reheating treatment, as we aimed to investigate the effects of reheating on the quality characteristics of pre-cooked meat products (PBC).

Response 1: Agree.

Comments 2: Regarding the sensory evaluation, we considered that PBC, as a prepared dish, might influence people's subjective assessments. Therefore, we used instruments such as an electronic nose and tongue to analyze its sensory parameters objectively. Comment: you can use GPT Chat to assess sensory evaluation or an electronic nose and tongue, but it is worth asking the panel for an assessment of curiosity. It means additional costs and time, but results could be better discussed. It is a suggestion for the future scientific work of the authors.

Response 2: Thank you for pointing this out. We agree with this comment. We also highly recognize the importance of sensory evaluation or the application of AI technology in sensory analysis for the outcome analysis of food research, and we intend to implement these methods in future studies. However, due to limitations in resources, these methods were not employed in this research.

Comments 3: The practical value of the experiment is that it provides valuable insights for consumers in selecting appropriate reheating methods for PBC, with potential applications for other pre-cooked meat products as well. The most preferable method for PBC reheating is steaming treatment (ST). Comment: the accents are correct for the importance of the experiment from a practical point of view.

Response 3: Agree.

Comments 4: In response 4, I am not sure what the author means in a sentence: Mainly, research neglects the impact of reheating treatment on pre-cooked dishes now.

Response 4: Thank you for pointing this out. Regarding this sentence, we meant to convey that the majority of research articles on meat do not focus on the effects of different reheating treatments on pre-cooked meat products. Therefore, we aim to supplement this area of research. We apologize for inadequacy in our expression.
